# Trajectory-conditioned reconstruction of single-cell expression suggests regulatory programs

**Wenjie Fan**[1,2,3,4], **Antonio Orvieto**[1,2,*] **Manfred Claassen**[3,4,5,6*]

[1]Max Planck Institute for Intelligent Systems, Tübingen, Germany
[2]ELLIS Institute Tübingen, Germany
[3]M3 Research Center, Faculty of Medicine, University of Tübingen, Germany
[4]Department of Internal Medicine I, University Hospital Tübingen, Germany
[5]Department of Computer Science, University of Tübingen, Germany
[6]Institute for Bioinformatics and Medical Informatics, University of Tübingen, Germany

`wenjie.fan@tuebingen.mpg.de, antonio@tue.ellis.eu`
`manfred.claassen@uni-tuebingen.de`

## Abstract

Foundation models for single-cell transcriptomics learn cell representations from millions of profiles, but are commonly pretrained on unordered cells and therefore do not explicitly condition on cell history. We introduce single-cell Transformer-iN-Transformer (scTNT), which conditions gene-expression reconstruction on inferred trajectories, represented here as ordered cell sequences. scTNT combines a frozen reduced-layer scGPT autoencoder with a trainable decoder-only transformer over sequences of latent cell embeddings and is trained by masked gene-expression reconstruction. On a CD8 T-cell exhaustion dataset with optimal transport-derived cell sequences, scTNT improves masked reconstruction relative to the scGPT baseline and outperforms alternative sequence backbones under controlled evaluations. We further propose a gradient-based gene-history attribution pipeline and apply TRRUST regulon enrichment to generate hypotheses about context-associated regulatory programs.

## 1 Introduction

Single-cell RNA sequencing (scRNA-seq) enables high-throughput measurement of gene expression across diverse cell populations (Hwang et al., 2018), yet most assays are destructive and yield snapshot observations, even when collected across time or experimental stages. As a result, modeling cell-state dynamics remains a central challenge: how cells progress along differentiation trajectories, branch into distinct fates, and how these transitions are driven by gene regulatory programs.

Large-scale models like scGPT (Cui et al., 2024) learn single-cell representations by pretraining on millions of transcriptomes, enabling scalable analyses of cell states and gene programs. However, these models are typically trained on unordered cells and encode cell states from single-cell inputs, leaving it unclear when and how conditioning on cell history improves representation learning or predictive performance beyond per-cell embeddings. A further open question is whether any history-dependent signal learned in this way supports meaningful biological interpretation.

In parallel, trajectory and pseudotime methods infer cell orderings from scRNA-seq data (Trapnell et al., 2014), and RNA velocity approaches such as scVelo estimate local transcriptional dynamics from spliced/unspliced counts to provide directionality along a process (La Manno et al., 2018; Bergen et al., 2020). These tools offer a practical proxy for cell history as ordered trajectories, but they do not by themselves provide a modeling framework for incorporating such history.

---

[*]These authors co-supervised the work.

Here we test whether conditioning on a proxy for cell history improves masked reconstruction beyond a representation learned from the current cell alone. In particular, we construct ordered cell sequences based on pseudotime and propose single-cell Transformer-iN-Transformer (scTNT), a modular approach that augments a frozen reduced-layer scGPT autoencoder with a trainable *context adapter* (a decoder-only transformer) operating over sequences of latent cell embeddings (Figure 1). scTNT is trained by masked reconstruction on short windows sampled from these sequences, enabling controlled comparisons to the corresponding frozen scGPT baseline. We evaluate on a CD8 T-cell exhaustion dataset with sequences constructed via optimal transport (OT) and report quantitative results (Figure 2). Finally, we probe whether the learned context signal is biologically structured using gradient-based gene-history attribution and TF regulon enrichment, generating hypotheses about context-associated regulatory programs (Figure 3).

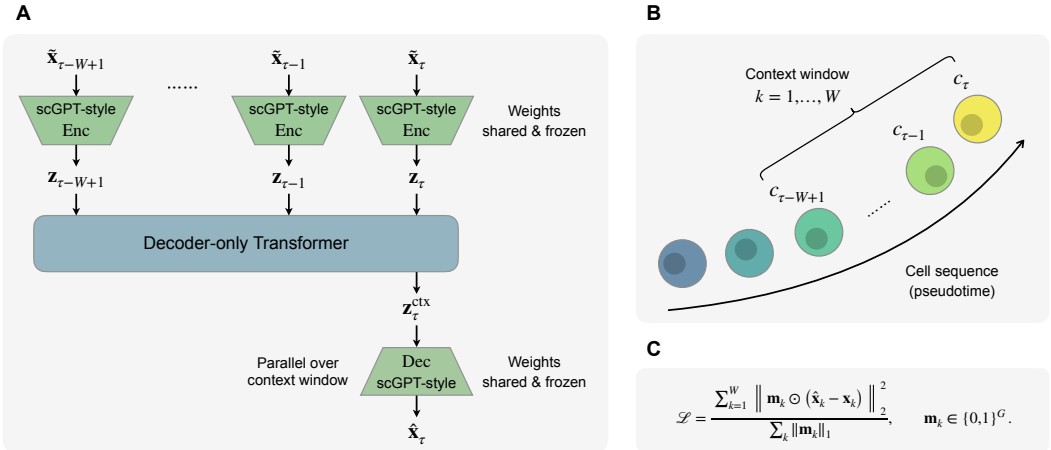

Figure 1: Overview of scTNT and the masked reconstruction objective. **(A)** Model: an scGPT autoencoder encodes each cell into a latent embedding; a transformer-based context adapter operates on the latent sequence and produces contextual latents, which the decoder maps back to gene space for reconstruction. **(B)** Context and window dataset: cells are ordered along pseudotime into sequences. For an endpoint $\tau$, we extract a length-$W$ window ending at $\tau$; within a window, slots are indexed by $k \in \{1, \ldots, W\}$ with $k \mapsto \tau - W + k$. **(C)** Objective: masked MSE computed only on masked gene entries across slots in the window.

Our contributions are:

- A latent-space decoder-only transformer on top of a frozen reduced-layer scGPT autoencoder to model history-dependent context for masked reconstruction.

- Evaluation protocols for cross-backbone comparisons and within-model context ablations.

- A gradient-based gene-history attribution pipeline with TF regulon enrichment to generate hypotheses about context-associated regulatory signals.

## 2 PROBLEM SETUP

**Data.** We use the T-cell exhaustion (TEX) dataset from Schleicher et al. (2025), comprising mouse CD8 T cells profiled by scRNA-seq over multiple collection time points during chronic lymphocytic choriomeningitis virus (LCMV) infection. We download the authors' released AnnData object, which contains the processed gene expression matrix and PCA coordinates, as well as cell-state annotations and scVelo-derived velocity pseudotime (visualized in Section A.1). We denote a cell by $c \in \mathcal{C}$, consisting of its gene expression and associated attributes used for sequence construction and interpretation (e.g., pseudotime and cell-state annotation). To match the scGPT autoencoder input, we select the top $G = 2000$ highly variable genes (HVGs) on the training cell split, defining the HVG set $\mathcal{G}$, and reuse $\mathcal{G}$ for the validation and test cell splits. We then follow the scGPT preprocessing workflow and discretize the expression values into 50 expression bins, plus a dedicated

zero-value bin. For binning, we use the scGPT binner implementation (Cui et al., 2024) and apply it separately per split. Let $\mathbf{x}(c) \in \{0, \ldots, 50\}^G$ denote the binned expression vector of cell $c \in \mathcal{C}$.

**Cell sequences.** We use the provided velocity pseudotime and discretize it into $T$ bins $\tau \in [T] = \{1, \ldots, T\}$. We define the inter-cell cost as the squared Euclidean distance in PCA coordinates after variance-normalizing each principal component. Let $\mathcal{C}_\tau \subset \mathcal{C}$ denote the cells in pseudotime bin $\tau$. For each adjacent pair $(\tau, \tau+1)$, we compute a one-to-one optimal transport (OT) assignment between equal-sized subsets of $\mathcal{C}_\tau$ and $\mathcal{C}_{\tau+1}$ using this cost. Chaining these pairwise assignments across $\tau = 1, \ldots, T-1$ yields cell sequences $\mathbf{c}^{(n)} = (c_1^{(n)}, \ldots, c_T^{(n)}) \in \prod_{\tau=1}^{T} \mathcal{C}_\tau$, indexed by $n \in [N]$, with $c_\tau^{(n)} \in \mathcal{C}_\tau$. We retain only full-length chains spanning all $T$ pseudotime bins; due to one-to-one matching at each step, each cell appears in at most one cell sequence.

**Splits.** Let $\mathcal{N}_{\mathrm{tr}}, \mathcal{N}_{\mathrm{va}}, \mathcal{N}_{\mathrm{te}}$ be a partition of $[N]$; this forms the training sequence split $\{\mathbf{c}^{(n)} : n \in \mathcal{N}_{\mathrm{tr}}\}$, and analogously for validation and test. In particular, this induces a training cell split, $\mathcal{C}_{\mathrm{tr}} = \{c_\tau^{(n)} : n \in \mathcal{N}_{\mathrm{tr}}, \ \tau \in [T]\}$, and analogously for validation and test. Under our OT construction, sequences are disjoint by construction, so the induced cell splits $\mathcal{C}_{\mathrm{tr}}, \mathcal{C}_{\mathrm{va}}, \mathcal{C}_{\mathrm{te}}$ are disjoint.

**Window dataset.** We use length-$W$ context windows sampled from cell sequences (Figure 1B) for training and for window-based evaluations that can be computed in parallel. Formally, we define the dataset of windows as:

$$\mathcal{D} = \Big\{ d = (n, e) \ \Big| \ n \in [N], \ e \in \{W, \ldots, T\} \Big\}.$$

Each window $d = (n, e) \in \mathcal{D}$ corresponds to the segment of sequence $n$ ending at global pseudotime bin $e$. This definition induces window splits $\mathcal{D}_{\mathrm{tr}}, \mathcal{D}_{\mathrm{va}}, \mathcal{D}_{\mathrm{te}}$ by restricting $n$ to $\mathcal{N}_{\mathrm{tr}}, \mathcal{N}_{\mathrm{va}}, \mathcal{N}_{\mathrm{te}}$. We index within-window positions by slot $k \in \{1, \ldots, W\}$ and define

$$c_{d,k} = c_{e-W+k}^{(n)}, \qquad \mathbf{x}_{d,k} = \mathbf{x}(c_{d,k}),$$

so that slot $k$ corresponds to global bin $\tau = e - W + k$. Unless needed, we omit $d$ and write $c_k$ and $\mathbf{x}_k$ for an arbitrary window.

**Masked reconstruction.** We train by masked reconstruction on length-$W$ windows. For a minibatch $\mathcal{B} \subset \mathcal{D}$ of size $B := |\mathcal{B}|$, indexed by $b \in \{1, \ldots, B\}$, each window provides binned expression $\mathbf{X}_b \in \{0, \ldots, 50\}^{W \times G}$ with rows $\mathbf{x}_{b,k}$ for slots $k \in \{1, \ldots, W\}$. We sample a binary mask $\mathbf{M}_b \in \{0, 1\}^{W \times G}$ and form the masked input by replacing binned expression at masked entries ($M_{b,k,g} = 1$) by $-1$ as in scGPT:

$$\tilde{\mathbf{X}}_b = (1 - \mathbf{M}_b) \odot \mathbf{X}_b + \mathbf{M}_b \odot (-\mathbf{1}).$$

Training loss is the mean squared error (MSE) computed only on masked entries (Figure 1C). The same masking and masked-MSE objective is used for all evaluations; evaluation protocols differ only in how the context for each target is selected (see Section 5.1). When minibatch indexing is not essential, we drop $b$ and write $c_k$, $\mathbf{x}_k$, and $\mathbf{X}$ for a single representative window.

## 3 METHOD

Given a cell sequence window $\mathbf{c} = (c_k)_{k=1}^{W}$ with masked gene-expression vectors $(\tilde{\mathbf{x}}_k)_{k=1}^{W}$, the frozen scGPT encoder maps each slot $k$ to a latent embedding $\mathbf{z}_k = \mathrm{Enc}(\tilde{\mathbf{x}}_k)$, yielding $\mathbf{Z} = [\mathbf{z}_1; \ldots; \mathbf{z}_W]$. The context adapter Ctx (a decoder-only transformer) treats cell latents as tokens and outputs contextualized latents, $\mathbf{Z}^{\mathrm{ctx}} = \mathrm{Ctx}(\mathbf{Z}) = [\mathbf{z}_1^{\mathrm{ctx}}; \ldots; \mathbf{z}_W^{\mathrm{ctx}}]$. Finally, a frozen scGPT decoder reconstructs each slot as $\hat{\mathbf{x}}_k = \mathrm{Dec}(\mathbf{z}_k^{\mathrm{ctx}})$. We compute for all $W$ slots in parallel.

**scGPT autoencoder.** The scGPT autoencoder consists of a transformer-based encoder Enc and a decoder Dec. For each cell, gene identities are treated as tokens and their binned expression values are provided as token-aligned inputs. We pretrain $\mathrm{Dec} \circ \mathrm{Enc}$ from scratch on the training cell split $\mathcal{C}_{\mathrm{tr}}$, then freeze the weights and train only the context adapter Ctx in latent space.

**Context adapter.** We model latent history dependence with a context adapter Ctx implemented as a decoder-only transformer with rotary positional embeddings. A causal attention mask ensures that at context slot $k$ the adapter attends only to slots $1, \ldots, k$. Because the scGPT encoder provides continuous latent embeddings per cell, Ctx operates directly in latent space and does not require discrete

tokenization. We decode directly from the final hidden states and therefore omit the language-model head for next-token prediction. In addition, we omit the final post-transformer LayerNorm to enable an identity initialization of the context adapter; see Section 4 for initialization details. For cross-model comparison, we replace Ctx with alternative sequence model backbones including an LSTM (Hochreiter & Schmidhuber, 1997), a per-channel AR, and the same transformer adapter trained with a shorter context length or a no-history control $W = 1$.

**Training objective.** Given a minibatch of $B$ windows drawn from $\mathcal{D}_{\mathrm{tr}}$ with targets $\mathbf{X} \in \{0, \ldots, 50\}^{B \times W \times G}$, reconstructions $\hat{\mathbf{X}} \in \mathbb{R}^{B \times W \times G}$, and masks $\mathbf{M} \in \{0, 1\}^{B \times W \times G}$, we minimize the masked MSE over masked entries:

$$\mathcal{L} = \frac{1}{\sum_{b,k,g} M_{bkg}} \sum_{b=1}^{B} \sum_{k=1}^{W} \sum_{g=1}^{G} M_{bkg} \big( \hat{X}_{bkg} - X_{bkg} \big)^2, \tag{1}$$

where $k$ indexes the $W$ slots within each sampled window.

# 4 IMPLEMENTATION DETAILS

We implement all models in PyTorch 2.1.2+cu121 and run experiments on Ubuntu 20.04 with an NVIDIA RTX 6000 GPU (48GB VRAM). Unless otherwise stated, we use context length $W=20$, sequence length $T=21$, and a mask ratio of $0.4$ for autoencoder pretraining, context-adapter training, and all evaluations. All cells are assigned the same scGPT batch label. In each training epoch and window-based evaluation run, we sample one length-$W$ window per sequence and compute losses/metrics over these sampled windows.

Our model combines (i) a reduced-layer scGPT autoencoder and (ii) a latent-space context adapter. We use the official scGPT package (Cui et al., 2024) to instantiate an autoencoder (2 transformer layers, 2 heads, latent dimension $64$) and train it from scratch on the TEX training split, keeping the study self-contained and computationally lightweight. The context adapter is a decoder-only transformer (2 layers, 2 heads, hidden dimension $64$) operating directly on the continuous latent embeddings. All context adapter backbones use the same hidden dimension and no dropout.

**Initialization.** We initialize all context adapter backbones to be an identity map so training starts from the frozen scGPT baseline. For transformer backbones, we zero-initialize the residual output projections in each block; all other weights are sampled from a zero-mean normal distribution with standard deviation $0.02$, and biases are initialized to zero. For the LSTM backbone, we use PyTorch's LSTM module and wrap it with skip connection $\mathbf{Z}^{\mathrm{ctx}} = \mathbf{Z} + \mathrm{proj}(\mathrm{LSTM}(\mathbf{Z}))$, zero-initializing the output projection $\mathrm{proj}$; other weights follow default initialization.

**Optimization.** For autoencoder training, we adopt the optimizer and mixed-precision setup used in scGPT fine-tuning (Adam without weight decay; AMP with gradient scaling) (Cui et al., 2024), but use a learning rate of $10^{-3}$ with cosine annealing for 1000 epochs. For context-adapter training, we freeze the autoencoder and train only the adapter using AdamW (learning rate $10^{-4}$; weight decay $10^{-3}$) with cosine annealing for 1000 epochs for all adapter backbones.

# 5 RESULTS

We report (i) quantitative results in Section 5.1 and (ii) attribution and enrichment analyses in Section 5.2. All evaluations are run on the test split and use the same masking procedure as training.

## 5.1 QUANTITATIVE RESULTS

We evaluate quantitative performance in two regimes. For $\tau$-resolved analyses (Figure 2A,B), we use a test sequence *scan-based* protocol that evaluates every target bin $\tau \in [T]$. For gene-level analyses (Figure 2C,D), we use a *window-based* protocol (one sampled length-$W$ window per sequence per masking replicate) for computational efficiency.

**Reconstruction gain.** We define reconstruction gain as the relative reduction in masked MSE compared to the scGPT autoencoder baseline. For any evaluation protocol producing masked losses

$\mathcal{L}_{\text{model}}$ and $\mathcal{L}_{\text{scGPT}}$ under the same masking scheme, we denote

$$\Delta\text{relMSE} = \frac{\mathcal{L}_{\text{scGPT}} - \mathcal{L}_{\text{model}}}{\mathcal{L}_{\text{scGPT}}}. \tag{2}$$

**Cross-model evaluation (scan-based).** We scan each full-length test sequence across targets $\tau \in [T]$. At target $\tau$, a model with context length $W$ receives the maximal available causal history, i.e., the target and its preceding $\min(W - 1, \tau - 1)$ bins, and we compute the masked MSE at $\tau$. Let $\mathcal{L}_{\text{model}}(\tau)$ and $\mathcal{L}_{\text{scGPT}}(\tau)$ denote the masked MSE aggregated over all test sequences and masked gene entries at fixed $\tau$. Applying Eq. 2 pointwise in $\tau$ yields a per-target gain profile $\Delta\text{relMSE}_{\text{scan}}(\tau)$ for cross-model comparison (Figure 2A).

**Context ablations (scan-based).** To assess context sensitivity of scTNT, for each target bin $\tau$ we keep the target cell fixed and compare $\Delta\text{relMSE}_{\text{scan}}(\tau)$ under the original history (*original context*) to the following ablations of the preceding $\min(W - 1, \tau - 1)$ bins: (i) *shuffle time*, which permutes cell order within the history; (ii) *no context*, which removes the history and evaluates the target alone; and (iii) *random context*, which replaces the history with cells sampled from $\mathcal{C}_{\text{te}}$ (Figure 2B).

**Gene-wise analysis (window-based).** For gene-level statistics, we evaluate on length-$W$ windows sampled from $\mathcal{D}_{\text{te}}$ and compute masked MSE as in Eq. 1. We form per-gene gains $\Delta\text{relMSE}(g)$ by restricting Eq. 1 to a single gene $g \in \mathcal{G}$ and aggregating across slots and sampled windows. These per-gene gains are used for the volcano plot and gene set enrichment analysis (GSEA) (Subramanian et al., 2005) in Figure 2C,D.

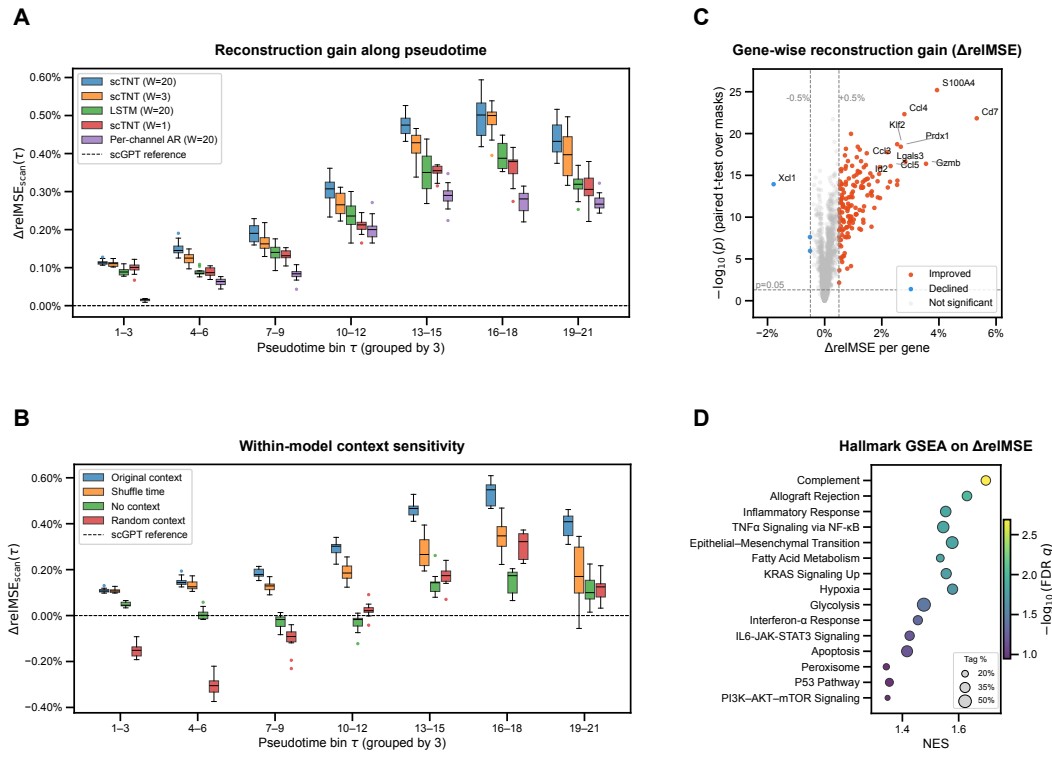

Figure 2: Quantitative evaluation of scTNT on the test split. **(A)** Cross-model comparison: $\Delta\text{relMSE}_{\text{scan}}(\tau)$ for scTNT and baseline models; boxplots show the distribution over masking replicates ($R = 10$). **(B)** Context sensitivity: $\Delta\text{relMSE}_{\text{scan}}(\tau)$ for scTNT under history ablations; boxplots show the distribution over masking replicates ($R = 10$). **(C)** Gene-wise gains: volcano plot of $\Delta\text{relMSE}(g)$ with paired $t$-test across masking replicates ($R = 30$); genes in the lowest 1% variance quantile across cells are excluded from the volcano plot; labeled genes show the strongest improvements or declines. **(D)** Hallmark GSEA: gene set enrichment by $\Delta\text{relMSE}(g)$, $g \in \mathcal{G}$; dot color encodes FDR $q$-value and marker size encodes tag fraction.

In Figure 2A, scTNT ($W{=}20$) achieves consistently positive scan-based reconstruction gain relative to the scGPT autoencoder across target bins $\tau \in [T]$, with the largest gains in mid-to-late bins $\tau \in [16, 18]$. Compared to shorter context ($W{=}3$), no-context control ($W{=}1$), and alternative backbones, scTNT ($W{=}20$) achieves higher $\Delta\text{relMSE}_{\text{scan}}(\tau)$, with the separation widening from early to mid bins $\tau \in [4, 15]$. Interpreting $\tau$ alongside cell state composition (Section A.2), these bins correspond to stages where cells undergo fate decisions towards T-cell memory-like versus exhausted states, consistent with larger benefits from inferred history.

In Figure 2B, ablating history reduces performance. Removing history (*no context*) collapses gains toward zero, indicating that improvements are not attributable to the autoencoder alone. Shuffling the history order (*shuffle time*) produces an intermediate drop, suggesting that temporal ordering carries information beyond marginal history content. Replacing history with random cells (*random context*) hurts performance most strongly at early $\tau$, compatible with scTNT leveraging coherent history rather than arbitrary cells.

In Figure 2C, gene-wise gains are heterogeneous. Despite modest aggregated $\Delta\text{relMSE}$, a subset of genes exhibits substantially larger relative gains. The strongest improvements include cytotoxic and effector-associated genes (e.g., *Gzmb*, *Ccl3*, *Ccl4*, *Ccl5*) (Chen et al., 2019), together with state-associated regulators/markers (e.g., *Id2* and the activation-associated marker *Lgals3*) (Masson et al., 2013; Smith et al., 2018). Significance is assessed by paired $t$-tests across masking replicates, pairing scTNT and scGPT gene-wise losses computed under the same masking replicate. Some genes show train/test split-dependent gain signs (e.g., *Xcl1*), with positive gains on train but negative gains on test, so we treat per-gene rankings as descriptive and prioritize enrichment-based summaries.

In Figure 2D, we rank genes by $\Delta\text{relMSE}(g)$, $g \in \mathcal{G}$ and run MSigDB Hallmark GSEA (Liberzon et al., 2015). Enriched terms highlight immune activation and stress-remodeling programs, including Complement, Allograft Rejection, Inflammatory Response, and TNF$\alpha$ signaling via NF-$\kappa$B, alongside Hypoxia and metabolic pathways (e.g., Glycolysis and Fatty Acid Metabolism). Interferon-$\alpha$ Response is also enriched, while IL6–JAK–STAT3 Signaling is weaker. This suggests that contextual reconstruction preferentially improves coordinated inflammatory, stress-associated, and interferon-response modules, motivating the TF regulon analysis in the next section.

## 5.2 Attribution and enrichment analyses

We next ask whether the learned context signal is biologically structured by attributing the reconstruction of a query gene to historical gene features and testing for enrichment of transcription factor (TF) regulons via the TRRUST v2 database (Han et al., 2018).

**Gradient attribution (window-based).** Fix a query gene $q \in \mathcal{G}$ and an evaluation slot $k_{\text{eval}} \in \{2, \dots, W\}$ within a sampled length-$W$ window. For each $b \in \{1, \dots, B\}$ in a minibatch $\mathcal{B} \subset \mathcal{D}_{\text{te}}$ and each history slot $k \in \{1, \dots, k_{\text{eval}}{-}1\}$, we compute the gradient magnitude of the reconstructed query expression with respect to the historical input feature for each gene $g \in \mathcal{G}$ (Figure 3A):

$$a_q(b, k, g; k_{\text{eval}}) := \left| \frac{\partial \hat{x}_{b, k_{\text{eval}}, q}}{\partial \tilde{x}_{b, k, g}} \right|. \tag{3}$$

We treat an attribution as valid only when (i) the historical feature is observed and (ii) the query gene at the evaluation slot is masked:

$$v_q(b, k, g; k_{\text{eval}}) := \mathbf{1}\{\tilde{x}_{b, k, g} \neq -1\} \cdot \mathbf{1}\{\tilde{x}_{b, k_{\text{eval}}, q} = -1\}. \tag{4}$$

We aggregate over history slots into a per-gene score at fixed $(b, k_{\text{eval}})$:

$$\bar{a}_q(b, g; k_{\text{eval}}) = \frac{\sum_{k=1}^{k_{\text{eval}}-1} v_q(b, k, g; k_{\text{eval}}) \, a_q(b, k, g; k_{\text{eval}})}{\max\left(1, \sum_{k=1}^{k_{\text{eval}}-1} v_q(b, k, g; k_{\text{eval}})\right)}. \tag{5}$$

and then average across batch elements and evaluation slots to obtain a context gene evidence score:

$$S_q(g) := \frac{1}{M} \sum_{b=1}^{B} \sum_{k_{\text{eval}}=2}^{W} \bar{a}_q(b, g; k_{\text{eval}}), \tag{6}$$

where $M$ is the number of included $(b, k_{\text{eval}})$ terms following the criterion in Eq. 4.

**Ablation-based context evidence.** Raw history attributions can contain signals that are not specific to *ordered* contextual information. To isolate context-specific evidence, we compare the original context to an ablated context where the history is shuffled and taken from another sequence (Figure 3B). Let $S_q^*(g)$ denote the context gene evidence score (Eq. 6) computed under the ablated context. We define a context-evidence score by subtraction:

$$\Delta S_q(g) \ := \ S_q(g) - S_q^*(g). \tag{7}$$

We use $\Delta S_q(g)$ as the ranking score for TF regulon enrichment.

**TRRUST TF scores and significance.** Let $\mathcal{F}$ be the set of TFs in TRRUST and $\mathcal{T}_f$ the target set (regulon) of TF $f$. We restrict to TFs with at least a minimum number of matched targets (e.g., $|\mathcal{T}_f \cap \mathcal{G}| \geq 5$). Given a query $q$, we score each TF by averaging ablation-based context evidence over its matched targets:

$$\Delta s_q(f) \ := \ \frac{1}{|(\mathcal{T}_f \cap \mathcal{G}) \setminus \{q\}|} \sum_{g \in (\mathcal{T}_f \cap \mathcal{G}) \setminus \{q\}} \Delta S_q(g), \tag{8}$$

where the query gene $q$ is removed from the TF target set to avoid trivial enrichment driven directly by $q$. To assess significance, we perform a permutation test: for each TF $f$, we sample random gene sets of size $|(\mathcal{T}_f \cap \mathcal{G}) \setminus \{q\}|$ from $\mathcal{G}$ to form a null distribution of TF scores, and compute a one-sided $p$-value. We correct across TFs using Benjamini–Hochberg and report FDR $q$-values (Benjamini & Hochberg, 1995).

For query genes, we choose the following markers spanning distinct CD8 T-cell programs: *Il7r* (memory-like) (Kaech et al., 2003), *Gzmb* (cytotoxic effector) (Sandu et al., 2020), *Ifit3* (interferon response) (Zhou et al., 2013), and *Havcr2* (TIM-3; exhaustion-associated) (Jin et al., 2010). We show TRRUST enrichment results for these queries in Figure 3C–F.

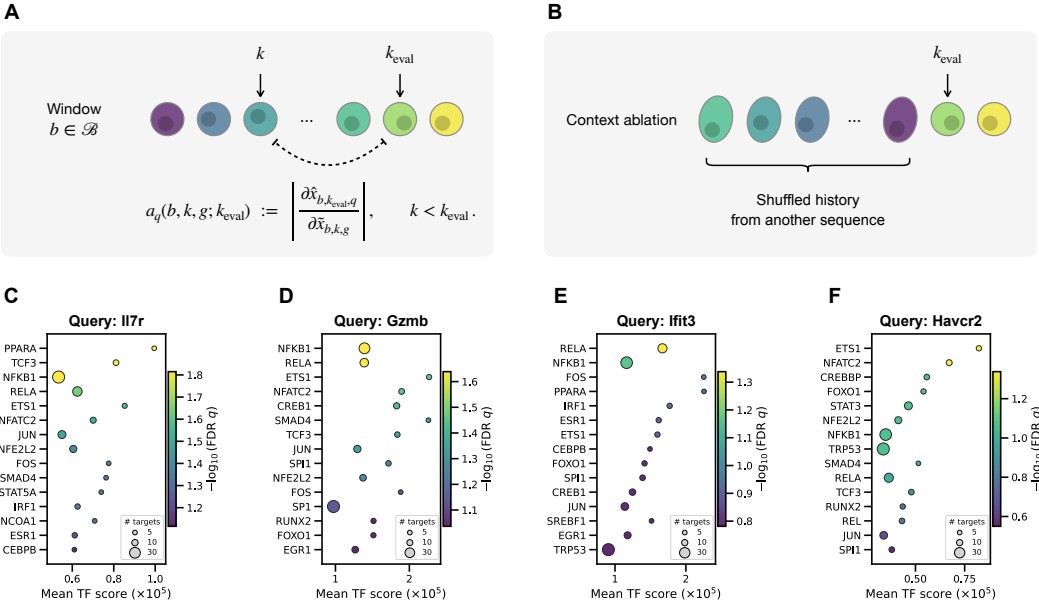

Figure 3: Gradient-based gene-history attribution on the test split. **(A)** Gradient attribution: for a query gene $q$ at evaluation slot $k_{\text{eval}}$, we compute $a_q(b, k, g; k_{\text{eval}}) = |\partial \hat{x}_{b,k_{\text{eval}},q} / \partial \tilde{x}_{b,k,g}|$ for history slots $k < k_{\text{eval}}$. **(B)** Context ablation: replace the history with an order-shuffled control drawn from another window within the same minibatch. **(C–F)** TRRUST TF enrichment results for selected query genes; x-axis shows TF scores $\Delta s_q(f)$, dot color encodes FDR $q$-value, and marker size indicates the number of matched TF targets.

Our attribution is magnitude-based and therefore does not encode directionality. Moreover, TF regulon enrichment should not be interpreted as direct regulation of the query gene; it tests whether

genes whose historical inputs most influence the query gene reconstruction are enriched for targets of a TF. We therefore treat these analyses as hypothesis-generating rather than causal.

Across queries, enriched regulons partially overlap but show query-dependent patterns. *Il7r* and *Gzmb* share enrichment of NF-$\kappa$B/AP-1–related factors, compatible with shared inflammatory/activation programs (Liu et al., 2017). In our analysis, *Il7r* shows relatively stronger enrichment of factors such as TCF3 (E2A) and PPAR$\alpha$, consistent with memory-associated regulatory and metabolic programs (Schauder et al., 2021; Saibil et al., 2019), whereas *Gzmb* shows relatively stronger enrichment of NFATC2 and CREB1, which have been implicated in effector/cytotoxic settings (Zhu et al., 2022; Kuijk et al., 2013). For *Ifit3*, we observe an interferon-linked signal (e.g., IRF1) alongside broader inflammatory factors, suggesting a mixed interferon/inflammation context (Schwartz et al., 2023). For *Havcr2*, the enriched TFs form a mixed profile; notably, NFATC2 has been reported to bind the *Havcr2* (TIM-3) promoter in LCMV-specific CD8 T cells (Zhu et al., 2022), though enrichment alone does not establish mechanism.

Overall, shared enrichments together with query-specific differences likely reflect a combination of common upstream programs in chronic infection and the coarse resolution of TF regulon enrichment on gradient-attribution scores. Stronger biological claims will require robustness checks (e.g., additional datasets, larger evaluation sets, and stability analyses across sampling seeds).

## 6    DISCUSSION

We introduced scTNT, which augments a frozen reduced-layer scGPT autoencoder trained from scratch on the TEX dataset with a latent-space context adapter operating over sequences of cell embeddings. Across quantitative evaluations on the held-out test split, incorporating a pseudotime cell history proxy improved masked reconstruction relative to the corresponding frozen scGPT baseline, with gains most evident at biologically meaningful positions. Gene-level analyses further suggested that the benefits are not uniform across genes, with improvements concentrating in coordinated immune activation/effector programs; hallmark enrichment offers a more stable summary.

Beyond quantitative results, we probed whether the learned context signal exhibits structure that maps to effector programs and antecedent TF-associated signals. Specifically, we evaluated the structure of the context signal via gradient-based gene-history attribution. By contrasting attributions under the original context with a shuffled-history control, we defined a context gene evidence score and tested for TF regulon enrichment using TRRUST. Enriched TFs partially overlap across query genes but also show query-dependent shifts that are coherent with distinct CD8 T-cell gene programs. While enrichment does not imply direct regulation of the query gene, it provides a compact, hypothesis-generating summary of context-associated regulatory signals.

This work has several limitations. First, results are shown on a single dataset with a specific proxy for cell history. An important next step is to validate scTNT across additional datasets and alternative sequence constructions that do not rely on optimal transport. Second, evaluations depend on sampled masking patterns. We mitigate this with multiple masking replicates, but larger evaluation sets would further improve the stability and resolution of gene and TF rankings. Third, our attribution is magnitude-based and thus omits directionality, and regulon enrichment is a coarse summary: overlapping regulons and shared upstream programs in chronic infection can yield similar TF signals across queries. Performing systematic stability analyses for TF enrichment and validating enrichment results with orthogonal evidence could strengthen the biological conclusions.

Overall, scTNT provides an adapter-style augmentation of a cell autoencoder that conditions latent-space reconstruction on inferred history from snapshot scRNA-seq data. Our initial attribution and enrichment analyses indicate coherent immune-related context signals, while motivating more rigorous validation across datasets and robustness settings.

## MEANINGFULNESS STATEMENT

Meaningful representations of cells should capture not only snapshot similarity but also dynamical progression and regulatory structure. Most single-cell foundation models learn static representations from unordered cells, without using temporal or lineage history. scTNT incorporates inferred cell history during self-supervised training to produce trajectory-conditioned latent representations

for gene-expression reconstruction. This encourages representation learning that captures differentiation dynamics rather than treating each cell independently. These context-aware representations enable the identification of genes whose reconstruction depends on prior context and highlight temporally associated regulatory programs, suggesting potential delayed regulatory effects and generating hypotheses about regulation along trajectories.

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

## A  APPENDIX

### A.1  TEX DATA

We visualize the cell-state annotations and velocity pseudotime in the TEX AnnData object (Schleicher et al., 2025) using the two-dimensional UMAP coordinates provided by the authors in Figure 4.

The provided CD8 T-cell state annotations include the following subtypes: early, proliferative, memory-like exhausted, intermediate exhausted, effector-like exhausted, and terminally exhausted. The scVelo-derived velocity pseudotime induces an ordering of cells along the differentiation process. In this work, we discretize the pseudotime into temporal bins and use it to construct ordered cell sequences for training and evaluation.

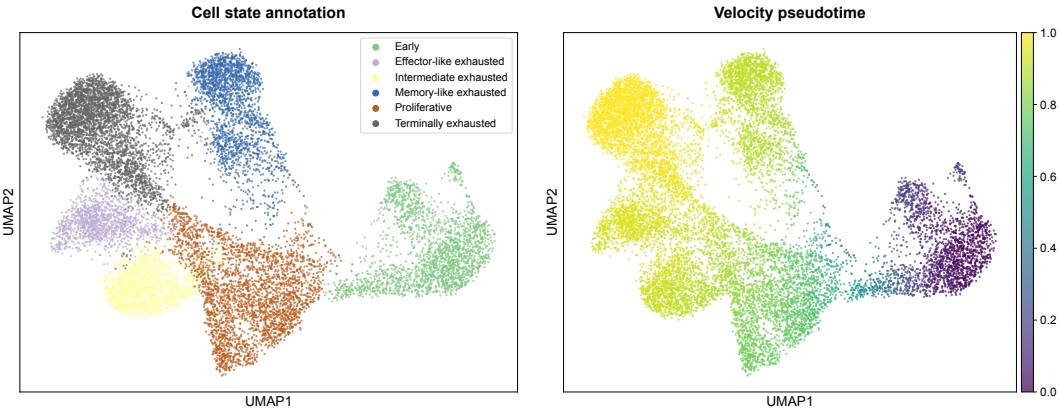

Figure 4: UMAP visualization of the TEX data. Left: provided CD8 T-cell state annotations. Right: scVelo-derived velocity pseudotime.

## A.2 INTERPRETING PER-TARGET RECONSTRUCTION GAIN ALONG PSEUDOTIME

We relate the per-target gain profile $\Delta\mathrm{relMSE}_{\mathrm{scan}}(\tau)$ of scTNT ($W = 20$) to the composition of annotated cell states across pseudotime bins $\tau$ in Figure 5. Mid-to-late bins $\tau \in [16, 18]$ exhibit highest gains in Figure 2A, and they correspond to a shift from effector-like exhausted states toward terminally exhausted states. A plausible interpretation is that this transition region contains trajectory-linked variation where consistent history reduces reconstruction ambiguity.

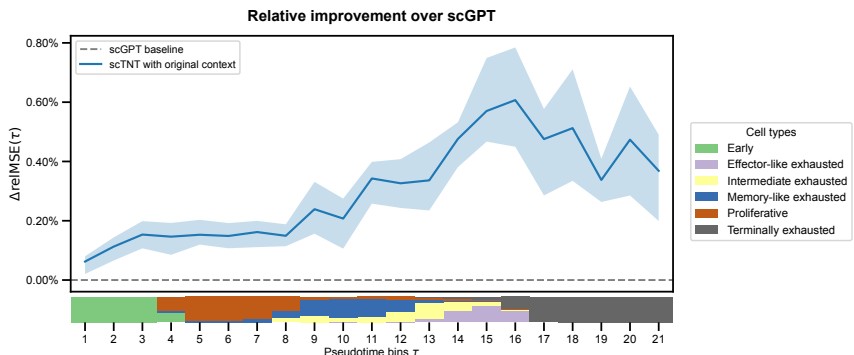

Figure 5: Per-target gain profile $\Delta\mathrm{relMSE}_{\mathrm{scan}}(\tau)$ with a reference strip showing the annotated cell-state composition across pseudotime bins in the TEX data. The line shows the mean across masking replicates and the shaded band shows the range (min–max) across replicates.

Bins $\tau \in [4, 15]$ span the region from proliferative to memory-like, intermediate, and effector-like exhausted states, where the state mixture becomes more heterogeneous and branching begins to emerge. In this region, the separation in $\Delta\mathrm{relMSE}_{\mathrm{scan}}(\tau)$ between backbones widens (Figure 2A), consistent with the interpretation that history becomes most beneficial when the current state remains compatible with multiple near-future continuations.

Finally, because the available causal history grows with $\tau$, gains at early bins are intrinsically limited by shorter contexts (regardless of model capacity). We therefore interpret $\Delta\mathrm{relMSE}_{\mathrm{scan}}(\tau)$ as the context benefit along the trajectory under the maximal available causal history at each $\tau$, rather than as a direct measure of "biological difficulty".

## A.3 USE OF LLMS

We made limited use of LLMs during the writing phase to improve the clarity and readability of the text. We did not use LLMs for ideation or to generate experimental results or analyses. All technical content, experiments, and conclusions are the work of the authors.

