# OpenReview forum: "Trajectory-conditioned reconstruction of single-cell expression suggests regulatory programs"
_ICLR.cc/2026/Workshop/LMRL — ICLR 2026 Workshop LMRL Poster_

### Official Review · Reviewer_nGRk · 2026-02-23

**Rating:** 6
**Confidence:** 4

**Review:**

Pros:
- The paper addresses an important and underexplored question of whether conditioning on inferred cell history improves single-cell foundation model performance, which is highly relevant to trajectory and dynamics modeling.
- The experimental design includes multiple evaluation regimes and context ablations, providing evidence that improvements are indeed due to temporal ordering rather than increased capacity.
- The paper is clearly written with strong figures and well-explained methodology, making the technical contributions easy to follow.


Cons:
- The empirical validation is limited to a single dataset, which significantly weakens claims about generality and biological relevance.
- Quantitative improvements appear modest, and the paper does not sufficiently analyze statistical significance or practical effect size beyond relative MSE reduction.
- The trajectory construction relies heavily on pseudotime and optimal transport assumptions, but sensitivity to trajectory inference errors is not evaluated.
- The model scale and backbone are relatively small compared to modern single-cell foundation models, raising questions about scalability and real-world applicability.
- Biological conclusions rely primarily on enrichment analyses without orthogonal validation, making them speculative rather than strongly supported.
- The interpretability method uses magnitude-only gradients, which limit mechanistic insight and may conflate correlated signals.
- The novelty is somewhat incremental, as trajectory conditioning and masked reconstruction have both been explored separately, and the main contribution is their combination rather than a fundamentally new modeling paradigm.

---

### Official Review · Reviewer_pmnJ · 2026-02-23
**Improve scGPT cell reconstruction with cell history**

**Rating:** 6
**Confidence:** 2

**Review:**

The authors of submission69 utilize pseudo-time information to improve scGPT reconstruction quality. The presented scTNT model is based on the scGPT model with an additional decoder transformer. The presentation is clear with detailed analysis of results as well as biological interpretations.
- The scTNT model achieves better reconstruction while the y-axis is in a 0.1% scale in Figures 2A and 2B, which implies that the absolute improvement is limited.
- In Figure 2C, about 3/4 of the genes show improvement in terms of relMSE regardless of significance. It is unclear why the remaining did not have any improvements, and also some of them are showing a significant decline.
- The pseudo-time information is calculated using scVelo. Will scTNT based on other pseudo-time algorithms also show similar improvements in reconstruction? In other words, how sensitive is the scTNT formulation?
- The authors complement scTNT results with enrichment analysis.

---

### Meta-Review · Area_Chair_JqtN · 2026-02-25

**Recommendation:** Accept (Poster)
**Confidence:** 4

**Metareview:**

Accept.

---

### Decision · Program_Chairs · 2026-03-02

**Decision:**

Accept (Poster)

**Comment:**

Please see the meta-review.